# The VaMa (Vatavuk and Marić) Artificial Intraocular Lens Capsule: A Novel Device and Method for Reversible Secondary Intraocular Lens Implantation in Patients with Aphakia Without Efficient Capsular Support

**DOI:** 10.3390/biomedicines13010162

**Published:** 2025-01-11

**Authors:** Goran Marić, Damir Godec, Bruno Krajačić, Marin Radmilović, Zoran Vatavuk

**Affiliations:** 1Department of Ophthalmology, UHC Sestre Milosrdnice, 10000 Zagreb, Croatia; radmilovic.marin@gmail.com (M.R.); zo.vatavuk@gmail.com (Z.V.); 2Faculty of Mechanical Engineering and Naval Architecture, University of Zagreb, 10000 Zagreb, Croatia; damir.godec@fsb.unizg.hr (D.G.); bruno.krajacic@fsb.unizg.hr (B.K.)

**Keywords:** artificial lens capsule, IOL exchange, IOL supporting platform, artificial intraocular lens supporting device, exchangeable IOL device, scleral fixation of IOL in aphakia

## Abstract

We describe a novel experimental device, the VaMa (Vatavuk and Marić) artificial intraocular lens (IOL) capsule, and a method that enables all IOL types to be implanted in the bag. We present the application of the device and the procedure in patients with aphakia and native capsule damage and without efficient capsular support. The VaMa device and the method facilitate IOL exchange due to refractive errors and, in the case of their invention, the implementation of superior IOLs in the future. The postoperative results after the implantation of the VaMa capsule along with IOLs in three patients are promising, with significant visual improvement and without adverse events 7 to 10 months postoperatively.

## 1. Introduction

The treatment of aphakia has evolved over the past decades. Conservative approaches, i.e., the use of glasses and contact lenses, can present cosmetic unacceptability, ring scotoma, aniseikonia, or poor compliance, and have led to the evolution of different surgical solutions [1], including many techniques for implanting an intraocular lens (IOL) [2]. However, none of these techniques are an universal solution for all aphakic eyes, and none have predominated in practice due to adverse events.

The indications for secondary IOL implantation have changed with the improved surgical outcomes of modern cataract surgery [1]. Phacoemulsification in noncomplicated cases has resulted in fewer adverse events, such as posterior capsule rupture (PCR), during the past decade [3]. The increased global aging of the population, as well as multiple ocular and systemic comorbidities—including diabetes, cataract, and concomitant antiVEGF therapy—seem to carry a higher risk of PCR [2,4]. In addition to PCR and zonular instability during cataract surgery, zonular deficiency primarily occurs in various eye traumas as well as in some diseases such as Marfan’s syndrome, homocystinuria, and pseudoexfoliation syndrome [5,6].

The visual expectations of patients after cataract surgery have increased over time and secondary IOL implantations, even as a part of IOL replacement surgeries, are, thus, performed. There are reasons for IOL exchange, such as dissatisfaction with multifocal IOLs, dysphotopsias [7], and, in some cases, outcomes with refractive error [8,9].

Various surgical possibilities such as sulcus implantation IOL, iris claw or suturing, transcleral fixation of IOL (TSFIOL), glued TSFIOL, Carlevale IOL, and the Yamane technique are available today [9], but all of them can bring their own adverse events, including uveitis-glaucoma-hyphema (UGH) syndrome, pupil ovalisation, endothelial cell loss, pigment dispersion syndrome, decentration and luxation of IOL, IOL tilt, intraoperative haptic fractures, difficulties related to IOL exchange if needed, target refraction errors [10,11,12], and opacification of the IOL due to air or gas for subsequent keratoplasties and vitrectomies followed by the gas tamponade [13,14].

One of the surgical solutions that solve most of these adverse events was described in 2021 by Balamurugan et al. as a platform for IOL implantation in an aphakic patient with inadequate capsular support [15], with a possibility of easily reversible IOL implantation if needed. However, this device seems to be designed only for three-piece IOLs with different refractive power and strictly positioned haptic orientation in the platform, making it insufficiently adaptable for toric IOL rotation.

The guidelines for choosing the type of surgery for secondary IOL implantation when adequate capsular support is absent remain controversial [16]. There are recommendations in ruling out certain conditions, such as a low endothelial cell number or iris defects, but the decision regarding the surgical type still mostly depends on the surgeon’s preferences and familiarity with the procedure [17,18].

In-the-bag implantation of the IOL remains the best way to position it if possible [8] and should be the lofty goal of secondary IOL implantation.

On the other hand, secondary IOL implantation into the capsular bag or exchange of the implanted IOL can be performed with the least trauma in the early postoperative period, before the formation of the capsular adhesions. In these cases, it is recommended to perform IOL implantation or IOL exchange within 4–6 weeks of the initial cataract surgery, although in-the-bag IOL exchange months after cataract surgery has been reported [9].

According to all of the above, the goals of secondary IOL implantation should include in-the-bag implantation, with the possibility of insertion of all the types of IOLs for that purpose, and also the possibility of rotating and achieving optimal positioning of the toric IOL. Furthermore, there should be a possibility of subsequent extraction of the IOL, for any reason, that is simple and causes as little trauma as possible, even years after the primary IOL implantation.

Well-known adverse events of modern surgical approaches for secondary IOL implantation, such as malposition of the IOL, unacceptable refractive power error, iris irritation, potentiation of UGH syndrome, and accelerated loss of corneal endothelial cells, should be avoided.

We present and describe a novel experimental device, the VaMa artificial intraocular lens capsule, and a method for IOL implantation in patients with aphakia who have natural lens capsule damage. The surgery was performed in three patients with different causes of aphakia, poor capsular support, and different additional ocular pathologies.

## 2. VaMa (Vatavuk and Marić) Artificial Lens Capsule—For Reversible IOL In-the-Bag Implantation

The device is made up of polydimethylsiloxane (medical grade PDMS) silicone elastomere and is designed in a universal single size of 10 mm for the circle’s outer diameter. The device’s shape can be adapted during the implantation through the corneoscleral incision, with preserved flexibility and shape memory. The wall is 0.2 mm at the thickest part, the height is up to 2 mm, and the weight is 20 mg. The peripheral edge of the rim has multiple miniature holes for the haptic of the artificial capsule or suture anchoring.

Figure 1 presents the various parts of the device. The device is divided into the main part (artificial lens capsule) and three or more auxiliary parts (sutures or artificial capsule haptics).

The body (1) has an anterior opening mimicking capsulorhexis, with a diameter of 8 mm (2); a posterior opening mimicking capsulectomy, with a diameter of 6 mm (3); the sulcus of the capsule for placing the IOL haptics (4); and multiple (at least three) equally distributed miniature holes for anchoring (5). Additionally, there are three or more sutures for flanging (6).

The device was created by the authors of this article as individuals and is a part of a modular implant invention under a European patent application (no. EP24168116.2).

## 3. Surgical Technique

The surgery is performed through several steps, as presented in the accompanying video (Appendix A).

The first step involves the execution of a 360° peritomy following the insertion of a 25 G (gauge) trocar through the sclera, as in the standard placement for a pars plana vitrectomy (PPV). A meticulous anterior vitrectomy with capsulectomy for capsular remnants is performed (Figure 2a and Figure 3a).

In the second step, the positions of the scleral fixation points for the artificial lens capsule are marked according to scheme in Figure 4, using the Casterviejo caliper at 12, 4, and 8 o’clock (Figure 2b and Figure 3b). After that, the main corneoscleral incision, 3.2 mm in width, is created at the superior 12 o’clock position (Figure 2c and Figure 3c). Viscoelastic (OVD) is injected in order to maintain the anterior chamber. In the next step, a 30 G needle is inserted in an almost tangential position, but parallel with the frontal (iris) plane, through the sclera on the previously marked pars plana position at 8 o’clock. Using the other hand and a Maxgrip forceps, the first haptic flanged prolene suture (Prolene 6-0, Ethicon, Bridgewater, NJ, USA), with one end previously flanged, is inserted through the main corneoscleral entry at the 12 o’clock position, with the unflanged end inserted first, and is placed in a needle lumen that is about 10 mm in length to avoid escape. The needle is externalized back through the sclera (Figure 2d and Figure 3d). The same step is repeated with the flanged suture in the 4 o’clock position (Figure 2e and Figure 3e). In the next step, a pair of McPherson forceps are used to roll the VaMa complex and push it lightly through the main corneoscleral incision, 3.2 mm in width, followed by the tightening of the sutures (Figure 2f and Figure 3f).

After that, the third prolene suture is positioned at 12 o’clock in the same way as the previous two at the 8 and 4 o’clock positions (Figure 2g and Figure 3g).

In the next step, the capsule is positioned in the centre and in the frontal plane by tightening the sutures and is held by the back side with an endoilluminator. The sutures are additionally tightened one by one, then cut and flanged on the outer end, right up to the surface of the sclera (Figure 2h and Figure 3h).

In the next step an IOL (Tecnis 1-Piece IOL, Johnson & Johnson, New Brunswick, NJ, USA) is injected in the eye and positioned in the artificial bag, just like into a natural lens capsule during standard phacoemulsification surgery, avoiding too fast injection, and the leading IOL haptic is positioned straight in the sulcus of the VaMa device. A chopper is used for rotating the IOL and the optimal positioning of the optic part and the second IOL haptic in the bag (Figure 2i and Figure 3i).

At the end, OVD is washed out of the anterior chamber and the corneoscleral suture is hydrated following the ceftriaxone injection in the anterior chamber. The pars plana trocars are pulled out and the sclerotomies are sewn by the resorbable sutures (Coated VICRYL 7-0, Ethicon, Bridgewater, NJ, USA), as well as the conjunctiva at two horizontal positions: 3 and 9 o’clock. Dexamethasone is injected subconjunctivally in a 1 mg dose, tobramycin ointment is applied locally, and the eye is finally covered with a cotton pad (Figure 2j and Figure 3j).

We performed the novel surgery on three patients within a 3-month period (Table 1). In the first patient, who had pseudoexfoliation syndrome as a risk factor, the reason for aphakia was an intraoperative rupture of the posterior lens capsule during phacoemulsification. The second patient had aphakia due to previous intraoperative complications during phacoemulsification (aqueous misdirection syndrome) and the expulsion of part of the iris. Both patients had the VaMa device implanted less than a month after the complications of phacoemulsification. The third patient had a previous penetrating eye injury involving the central part of the cornea, as well as injury to the iris and the lens. Immediately after the injury, about a year before VaMa implantation, this patient was primarily surgically treated with corneal suturing, lensectomy, and PPV. All three patients were examined frequently in the first postoperative weeks after VaMa implantation surgery. Their recovery was rapid and without adverse events. Anti-inflammatory therapy was included (initially dexamethasone drops six times a day, ointment three times a day), with a gradual dose reduction from the end of first postoperative week to the end of fourth postoperative week. One month after VaMa implantation surgery, the patients were examined once a month for a total follow-up period of 7 to 10 months.

Postoperative follow-up examinations included a best corrected visual acuity (BCVA) test, a slit lamp exam, measurement of intraocular pressure (Goldmann aplanation tonometry), corneal topography, specular microscope imaging, anterior OCT (for iridocorneal angles), and OCT imaging of the macula and optic nerve head (ONH). Ultrasound biomicroscopies (UBMs) give us enough information on the basis of which we can conclude the position of the implant, but they are not of sufficient quality to be included. There were no postoperative signs of inflammation or increases in the secondary intraocular pressure. BCVA was improved in all patients with achieved target refraction. Minimal myopia was present, equalling up to −0.50 diopters of the spherical equivalent. Specular microscopy showed that there was no significant corneal endothelium cell loss. Anterior OCT showed that there was an open iridocorneal angle, without contact between the posterior surface of the iris and the VaMa implant in the full 360° circumference. OCT of the macula and the optic nerve head (ONH) were without any change postoperatively.

The first patient also underwent magnetic resonance imaging (MRI), which showed the postoperative state of the VaMa and IOL implanted in the eye (Figure 5), where the left eye with the natural lens cannot be clearly distinguished from the right eye with the complex VaMa artificial capsule and IOL. This position of the IOL in aphakic eyes is closest to the physiological one.

## 4. Discussion

Every artificial implant should strive for the best possible imitation of the native tissue it replaces, and be as close as possible in terms of anatomy and function. In addition, it should be as easy to implant as possible, but also as postoperatively stable as possible.

Accordingly, the VaMa capsule is made of medical grade silicone for long term implantation in the human body. Due to its chemical properties, as well as its design performance and its physical properties in this shape and size, it seems to be harmless to the surrounding tissue. It is suitable for intraoperative manipulation too, after which it returns to its original state due to its material shape memory.

### 4.1. No Iris Contact to Avoid Anterior Chamber Adverse Events

The VaMa artificial capsule is made in one size, and its application in eyes of extremely different axial lengths can be optimized by fitting capsule haptics that are of a custom length. These dimensions and flexibility are sufficient to keep the corneoscleral incision no wider than 3.2 mm, which is important for avoiding large surgically induced astigmatism (SIA).

Given that the anterior OCT showed an open iridocorneal angle, there was no contact between the posterior surface of the iris and the VaMa implant in the full 360° circumference, and there was no tilt, which is certainly useful for avoiding adverse events in the anterior segment of the eye. UBM with higher quality images for accurate representation of the VaMa capsule position of the artificial lens should be performed in further studies.

### 4.2. VaMa Fixation to the Sclera

Implantation of the VaMa artificial capsule using the technique of flanged polypropylene sutures was intended as a temporary method of attachment. However, it proved to be satisfactory in terms of intraoperative handling and efficiency of IOL positioning, with no postoperative target refraction errors. The recovery of all patients in our series was quick and without complications.

The follow-up period for our patients (7 to 10 months) was too short to record the eventual conjunctival erosion of the flanged end sutures placed subconjunctivally, as recorded for other techniques [19,20]. The possible appearance of erosion is definitely planned to be solved by an upgrade in the form of haptics for the artificial capsule, as contained in the above-mentioned patent application for the VaMa device.

The distance of the scleral fixation positions from the limbus is marked according to Figure 4. The starting point for the calculation is at a distance of 3 mm from the horizontal limbus positions at 3 o’clock and 9 o’clock, which have previously been shown to be effective in terms of the predictability of postoperative refraction for secondary IOL implantation and positioning [21,22,23]. There were three fixation points, spaced 120° apart, in order to achieve the desired plane and avoid tilt. The scleral fixation points were calculated by considering a circle on the pars plana projection to the sclera, along which the anchors should be located, equally distant from the centre of the optical axis and the cornea, and considering that the cornea is an ellipse with a longer horizontal and shorter vertical axis [24]. Accordingly, and keeping in mind the goal of positioning the VaMa in the frontal plane, the distances from the limbus are not equal for each position.

### 4.3. IOL Power Choice and Target Refraction

This technique had satisfactory and expected target postoperative refraction in all three patients in our series. Minimal myopia is present. Nevertheless, the predictability of the target refraction with this positioning should be tested using a larger sample. In accordance with publications that show no significant difference between the IOL formulas, the SRK/T, Holladay II, and Barret 2 formulas were used for the IOL refractive power choice [21,23], as is standard for in-the-bag IOL implantation.

However, this positioning of the artificial capsule may also require a special approach to calculating IOL power in the future.

### 4.4. IOL Exchangeability

Considering that adhesions do not occur between the two artificial materials, the IOL and the VaMa, it is possible to replace the IOL fairly atraumatically for any reason, even after a long time. This may be most important in paediatric cataract surgery, where possible late refractive errors may occur due to the lack of real anticipation of the postoperative growth of the child’s eye [21,25,26,27]. This opens the possibility for a paradigm shift in the treatment of children’s cataracts. Further investigations for that indication are yet to be done.

### 4.5. Additional Considerations

This device and method open up a number of possibilities in solving aphakia of various etiologies and the conditions associated with it, such as iris defects and secondary glaucoma. There is a possibility that such an implant could be a platform for fixing an artificial iris, serve as a drug depot, and aid not only IOL implantation, but even mini-telescope implantation. Furthermore, the haptics for the artificial capsule could be MIGS/MIBS stents, as covered by our European patent application.

## 5. Conclusions

We demonstrated that the implantation of the novel VaMa artificial lens capsule along with IOLs was successfully performed in three patients without any adverse events at 7–10 months post-operation. There were no refractive unexpected outcomes. The postoperative results are promising, with significant visual improvement, and show many potential advantages of the novel VaMa device and method compared to previous methods of secondary IOL implantation. Further studies involving larger samples and a longer follow-up period are needed to confirm the safety of the VaMa device and method.

## Figures and Tables

**Figure 1 biomedicines-13-00162-f001:**
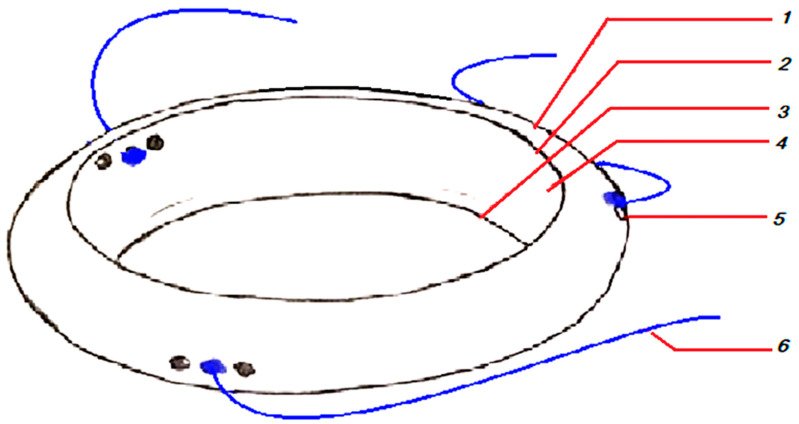
The structure of the VaMa device: 1—artificial lens capsule body; 2—anterior opening; 3—posterior opening; 4—sulcus for IOL haptics; 5—anchoring holes; 6—sutures for double flanging.

**Figure 2 biomedicines-13-00162-f002:**
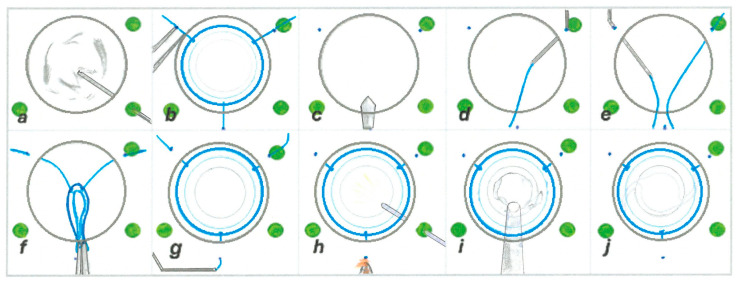
(**a**–**j**): Diagrammatic representation of the surgery (surgeon’s superior view). Subfigure (**a**) shows the 360° peritomy followed by adequate cauterization of the blood vessels, the insertion of 25 G trocars for pars plana vitrectomy, and a meticulous anterior vitrectomy with capsule remnant capsulectomy. (**b**) The VaMa device is put on the corneal surface with three one-side flanged polypropylene sutures 6-0 to help mark the scleral positions for VaMa anchoring with the limbus distance; according to the scheme in Figure 4, it is 3.4 mm at 12 o’clock, and 3.1 mm at 4 and 8 o’clock. (**c**) Performing the main corneoscleral incision, 3.2 mm in width at 12 o’clock position, is followed by viscoelastic injection in the anterior chamber. Subfigure (**d**) shows the insertion of the 30 G needle at 8 o’clock through the sclera, in parallel with the iris, and the insertion of the polypropylene suture through the corneoscleral entrance using Maxgrip forceps. It is put into the needle lumen, which is about 10 mm in length, followed by needle externalisation together with the suture at the position 8 o’clock. (**e**) The same step is repeated at the 4 o’clock position. (**f**) The VaMa is folded using McPherson forceps and then inserted into the anterior chamber; alternately, the surgeon is almost simultaneously pulling the polypropylene sutures apart. (**g**) The step before the previous one is repeated, now with the 30 G needle and the polypropylene suture at the 12 o’clock position with additional tightening, leading the VaMa to the optimal position. (**h**) An endoilluminator is used to hold the VaMa in the right frontal position and all of the three sutures are tightened and flanged on the outer end, right up to the surface of the sclera. (**i**) A one-piece IOL is injected through the main port and placed in the bag of the VaMa. (**j**) The viscoelastic is washed out, and the corneoscleral paracentesis ports and anterior chamber are hydratisated with BSS and ceftriaxone, followed by the closure of the trocar inlets, a peritomy with resorbable sutures, and the application of a dexamethasone subconjunctival injection.

**Figure 3 biomedicines-13-00162-f003:**
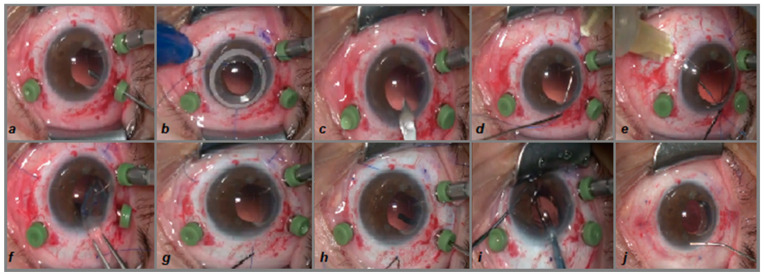
(**a**–**j**): Important steps of the surgery (surgeon’s superior view). Subfigure (**a**) shows a 360° peritomy followed by the insertion of 25 G trocars for a pars plana vitrectomy and a meticulous anterior vitrectomy and capsulectomy. Subfigure (**b**) shows the marking of the scleral positions for VaMa anchoring, according to scheme in Figure 4. (**c**) The main corneoscleral incision, 3.2 mm in width at 12 o’clock position, is made. Subfigure (**d**) shows the insertion of the 30 G needle at 8 o’clock through the sclera and in parallel with the iris, and the insertion of the polypropylene suture through the corneoscleral 12 o’clock entrance using Maxgrip forceps. It is then put into the needle lumen, which is about 10 mm in length, followed by needle externalisation together with the suture at the 8 o’clock position. (**e**) The previous step is repeated at the 4 o’clock position. (**f**) The VaMa is folded and inserted through the main corneoscleral incision into the anterior chamber. Subfigure (**g**) shows the insertion of the 30 G needle at the 12 o’clock scleral position and the polypropylene suture through the main corneoscleral incision into the lumen of the needle with additional tightening, leading the VaMa to the optimal position. (**h**) An endoilluminator is used to hold the VaMa in the right position, and all of the three sutures are tightened and flanged on the outer end. (**i**) A one-piece IOL is injected and placed in the bag of the VaMa. (**j**) The viscoelastic is washed out and the corneoscleral paracentesis ports and anterior chamber are hydratisated with BSS and ceftriaxone, followed by closure of the trocar inlets, a peritomy with resorbable sutures, and the application of a dexamethasone subconjunctival injection.

**Figure 4 biomedicines-13-00162-f004:**
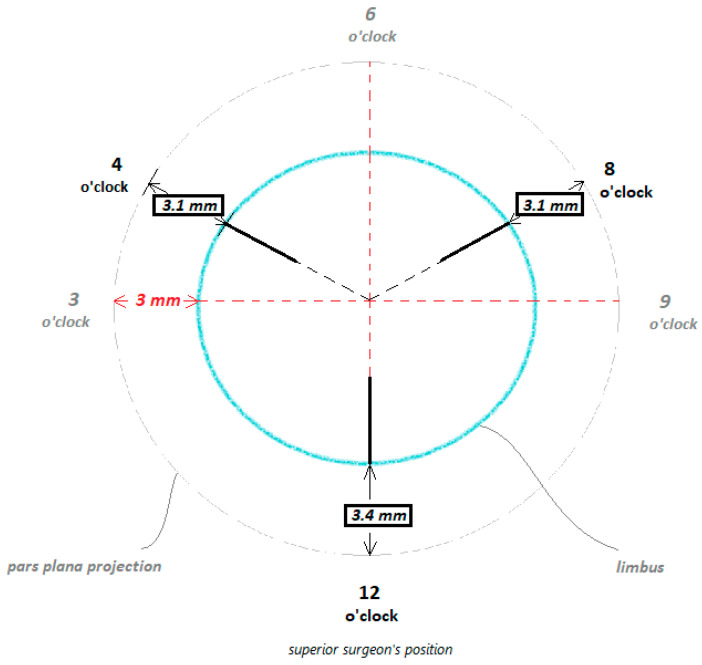
The calculation scheme for marking the limbus distance of the scleral anchoring sutures, depending on their positions. The sutures are about 3.1 mm at the 4 o’clock and 8 o’clock pars plana positions and about 3.4 mm at the 12 o’clock position.

**Figure 5 biomedicines-13-00162-f005:**
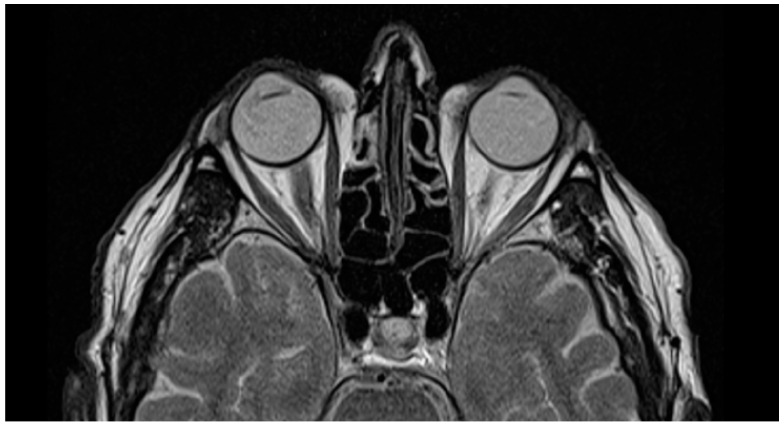
MRI shows the VaMa implanted in the right eye of the first patient.

**Table 1 biomedicines-13-00162-t001:** Presentation of three patients implanted with VaMa artificial intraocular lens capsule.

Patient No.	Age, Sex	Eye	Cause for Aphakia/Poor Capsular Support; Other Pathologies	Primary Surgical Intervention	Secondary Surgical Intervention	AL, WTW (mm)	Preoperative BCVA (Decimal) / IOP (mmHg) / AG Therapy	Follow-Up (Months)	Postoperative VA and BCVA (Decimal) / IOP (mmHg) / AG Therapy
1	71 y, M	R	Cataract surgery, PCR; PEX	PHACO + VAP	PPV + VaMa + IOL	24.55, 11.6	0.6 / 22 mmHg / drops: timolol, dorzolamide	10	VA = 0.9 BCVA (−0.50 D sph) = 1.0 / 14 mmHg / ***No*** drops
2	70 y, F	R	Cataract surgery, aqueous misdirection syndrome and iris prolapse; dry eye syndrome	PHACO + VAP, corneoscleral suture	PPV + VaMa + IOL	23.00, 12.0	0.8 / 16 mmHg / No drops	8	VA = 0.8 BCVA (−0.75 D cyl ax 170°) = 1.0 / 15 mmHg / No drops
3	62 y, M	R	PPV, lensectomia, corneal suturing due to previous penetrating eye injury	PPV, lensectomia, corneal sutures	PPV + VaMa + IOL	23.42, 10.8	0.025 / 24 mmHg / drops: timolol, dorzolamide, brimonidine	7	VA = 0.1 Not corrected (due to irregular astigmatism) / 18 mmHg / drops: timolol, dorzolamide, brimonidine

Abbreviations: y—years, M—male, F—female, R—right eye, PCR—posterior capsule rupture, PEX—pseudoexfoliation syndrome, PHACO—phacoemulsification cataract surgery, VAP—partial anterior vitrectomy, PPV—pars plana vitrectomy, VaMa—Vatavuk and Marić artificial lens capsule, IOL—intraocular lens, AL—axial length of the eye, WTW—“white to white” horizontal corneal diameter, VA—uncorrected visual acuity, BCVA—best corrected visual acuity, D sph—sphere diopter, D cyl—cylinder diopter, IOP—intraocular pressure, AG—anti-glaucoma.

## Data Availability

The original contributions presented in this study are included in the article/Appendix A. Further inquiries can be directed to the corresponding author(s).

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
