# Peer review of "The VaMa (Vatavuk and Marić) Artificial Intraocular Lens Capsule: A Novel Device and Method for Reversible Secondary Intraocular Lens Implantation in Patients with Aphakia Without Efficient Capsular Support"

_biomedicines, 2025, doi:10.3390/biomedicines13010162_

Round 1

Reviewer 1 Report

Comments and Suggestions for Authors

The authors present a very interesting novel medical device for assisting secondary intraocular lens implantation in situations with inadequate capsular bag support without specialized sulcus intraocular lenses.
The methods and device design seem to be clear but there are numerous questions to be answered and the manuscript could be improved by adding some more data:

1) The authors should disclose where the device was manufactured, whether the manufacturer was registered for the manufacture of medical devices according to EU regulation 745/2017 (MDR)

2) What's the regulatory status of the device? Did the authors also check for clearance from the competent authority in their country according to MDR Art. 62 and 82?

3) The device is not really visible in the video. The authors state that anterior segment OCT has been performed in the patients. Please add postoperative images (OCT or UBM), so that the extent and position of the device can be evaluated

4) Any follow-up study should also evaluate the required power correction for the IOL calculation and the possible IOL decentration with the device. Was any correction applied for the power of the IOLs implanted?

5) Please add the postoperative refraction and uncorrected VA

6) Could a 10 mm ring be too small for myopic eyes with large pupil diameter? 

Author Response

Dear Reviewer,

Thank you very much for taking the time to review this manuscript.

Please find the detailed responses below and the corresponding revisions/corrections highlighted/in track changes in the re-submitted files.

Comments 1:

The authors should disclose where the device was manufactured, whether the manufacturer was registered for the manufacture of medical devices according to EU regulation 745/2017 (MDR).

Response 1:

The artificial lens capsule (VaMa) device was produced as a collaboration between doctors from Ophtalmology department of UHC Sestre milosrdnice, Zagreb, and engineers from the Faculty of Mechanical Engineering and Naval Architecture, University of Zagreb, Croatia (all authors are included in the article).

Comments 2:

What's the regulatory status of the device? Did the authors also check for clearance from the competent authority in their country according to MDR Art. 62 and 82?

Response 2:

The device VaMa is currently in the experimental phase and the commercialization process has not been initiated. All patients were informed about this before signing the informed consent, as well as the ethics committee of the UHC Sestre milosrdnice, which approved the clinical research. So, at (this) moment of submission of this article for publication, it is not yet subject to the regulations of the MDR.

Comments 3:

The device is not really visible in the video. The authors state that anterior segment OCT has been performed in the patients. Please add postoperative images (OCT or UBM), so that the extent and position of the device can be evaluated.

Response 3:

The device is best seen in Figure 3b and in Video S1 at approximately 40–45 seconds when unfolded, as well as 75–100 seconds when implanted. It has been described in detail previously (page 3, Figure 1), so it is not difficult to follow the positioning and surgical technique of implantation. The performed anterior OCT and especially the UBM give us enough information on the basis of which we can conclude the position of the implant, but they are not of sufficient quality or representative images to be included in the article. This is explained on the page 8, paragraph 1, lines 1-3 and page 9, paragraph 1, lines 4-5 and described as a deficiency of the study. That is why we performed the MRI. We will definitely try to get better quality recordings in our further work on this study.

Comments 4:

Any follow-up study should also evaluate the required power correction for the IOL calculation and the possible IOL decentration with the device. Was any correction applied for the power of the IOLs implanted?

Response 4:

We are aware of the challenges related to the calculation of IOL power and aiming the target refraction. In further research on a larger sample of patients, it will be easier to draw correct conclusions. This is already stated on the  page 9, paragraph 4.3. (IOL Power Choice and Target Refraction).

Comments 5:

Please add the postoperative refraction and uncorrected VA.

Response 5:

Postoperative uncorrected visual acuity (VA) and refraction for best-corrected visual acuity (BCVA) are added in the Table 1.

Comments 6:

Could a 10 mm ring be too small for myopic eyes with large pupil diameter?

Response 6:

These dimensions of the artificial capsule (ring) seem suitable for all eyes, regardless of the axial length of the eye. The required size of the artificial capsule primarily depends on the White-To-White (WTW) dimensions which are not linear as can be seen in the ocular biometry literature (1), “Notably, the association between WTW distance and AL was not linear. As the AL increased, the WTW distance initially increased, reached a peak in the group with ALs of 24.5 to 26 mm, and then slowly decreased. However, all of the myopic eyes (AL > 24.5 mm) still had larger WTWs than the normal and short eyes (AL ≤ 24.5 mm).” and “The mean WTW distance was 11.69 ± 0.46 mm.” 

The size and diameter of the pupil is more influenced by the regulation of the autonomic nervous system (2, 3) than by eye biometry (axial length). The pupil is almost never physiologically wider than 8 mm, which is dimension of the anterior opening of the artificial capsule, and the optical part of most IOLs for in-the-bag implantation is 5-6 mm and the dimension of the posterior opening of the artificial lens capsule is 6 mm.

In accordance with the above, it will probably not be necessary to produce a custom ring size of the artificial capsule.

What will probably need to be adjusted individually for each patient is the length of the each haptic (in the article – flanged prolene suture) of the capsule itself and the distance of their position from the limbus due to the difference of pars plana width in the eyes with different axial length (4).

References (only for correspodence with the Reviewer)

  1. Wei L, He W, Meng J, Qian D, Lu Y, Zhu X. Evaluation of the White-to-White Distance in 39,986 Chinese Cataractous Eyes. Invest Ophthalmol Vis Sci. 2021 Jan 4;62(1):7. doi: 10.1167/iovs.62.1.7. PMID: 33393973; PMCID: PMC7794278.,

  1. Bitsios P, Prettyman R, Szabadi E. Changes in autonomic function with age: a study of pupillary kinetics in healthy young and old people. Age Ageing. 1996 Nov;25(6):432-8. doi: 10.1093/ageing/25.6.432. PMID: 9003878.

  1. Li K, Li X, Wang Q, Wang L, Huang Y. Kinetic pupillary size using Pentacam in myopia. Front Neurosci. 2022 Nov 25;16:981436. doi: 10.3389/fnins.2022.981436. PMID: 36507361; PMCID: PMC9732367.

  1. Sudhalkar A, Chauhan P, Sudhalkar A, Trivedi RH. Pars plana width and sclerotomy sites. Ophthalmology. 2012 Jan;119(1):198-9.e1-3. doi: 10.1016/j.ophtha.2011.09.002. PMID: 22214943.

Best regards,

Goran Marić and Co-Authors

Reviewer 2 Report

Comments and Suggestions for Authors

The study addresses a very innovative topic presenting a new device for the surgical treatment of aphakia. Aphakia is a condition that requires a personalized surgical approach based on the clinical condition of the eye and this study illustrates a new technique that adds to those already present. The device can be very useful in some selected cases and in my opinion this article highlights its advantages and disadvantages and above all the implantation methods. The descripted device allows for the implantation of IOLs when the capsular bag is absent or severely damaged and can represent a valid therapeutical option. The study illustrates 3 very different clinical cases highlighting the versatility of the procedure. The article is explanatory and very detailed in illustrating the surgical technique and the supplementary video material offers further help. In my opinion the article can be published without particular changes

Author Response

Dear Reviewer,

Thank you very much for taking the time to review this manuscript.

We really appreciate that.

Best regards,

Goran Marić and Co - Authors

Round 2

Reviewer 1 Report

Comments and Suggestions for Authors

The authors reply was sufficient and required changes were included